# The Bike-Sharing System in Slovakia and the Impact of COVID-19 on This Shared Mobility Service in a Selected City

Stanislav Kubaľák [ID], Alica Kalašová [ID] and Ambróz Hájnik *

Department of Road and Urban Transport, University of Žilina, 01 026 Žilina, Slovakia;
stanislav.kubalak@fpedas.uniza.sk (S.K.); alica.kalasova@fpedas.uniza.sk (A.K.)
* Correspondence: ambroz.hajnik@fpedas.uniza

**Abstract:** The current COVID-19 pandemic situation has a very significant impact on urban mobility, as evidenced by fundamental changes in passengers' behavior. In many cases, passengers have switched to other modes of transport to minimize the risk of infection (particularly cycling and walking). This paper aims to point out the analysis results of the impacts of a bike-sharing system in the city of Košice before and during the COVID-19 pandemic. Additionally, this paper describes an analysis of the current state of bike-sharing in the Slovak Republic. We have stated a timeline of significant dates related to the COVID-19 pandemic. COVID-19 had a significant impact on people's mobility and bike-sharing, as evidenced by the graphs and results in this paper.

**Keywords:** shared economy; mobility; bike-sharing; COVID-19; Slovakia

## 1. Introduction

At present, the car is the most commonly used means of transportation. This often leads to major problems in cities, such as congestion, emissions, traffic accidents and long delay times. Transport is an important area that directly affects the country's economy [1].

Most large cities address heavy traffic and a lack of parking spaces by investing in the development of public transport and new forms of mobility [2]. One of the most suitable solutions is to switch passengers from cars to public transport vehicles and other transport modes (e.g., walking and cycling). The attractiveness of bike-sharing and public transport can affect passengers and the number of cars on the road network. It is possible to increase passengers traveling via public transportation by upgrading transport vehicles; however, sufficient funding is needed to increase its attractiveness and competitiveness [3]. Along with the development of public transport, it is appropriate to introduce a bike-sharing system. Thanks to that, passengers can travel to places where there is no public transport route.

Every year, there has been an increase in published scientific articles on the topics of both bike-sharing and how the pandemic has impacted urban mobility [4].

Bike-sharing is used mainly in cities, where it facilitates the last mile of transportation among commuters, meaning transport to places where public transport has no route [5]. The bike-sharing system can also be used by companies in the form of the use of free space for advertising products and services [6].

The target group of bike-sharing is users with the need to travel to a specific place. For example, the user can be a tourist who wants to see and visit the surroundings of the city, a student or worker who needs to be transported to school or work or residents who do not own a bicycle or do not have bicycle storage facilities [7]. A great advantage for bike-sharing operators is the possibility of achieving revenue from advertising space [8].

The advantages for bike-sharing users are the low price and availability. Furthermore, bike-sharing supports the reduction of the number of cars on the roads and thus helps to improve traffic in cities. This reduces accidents and increases traffic safety. Bike-sharing is

also environmentally friendly and has a positive effect on human health. Thus, it is a more sustainable mode of transport compared with public transport [9].

The disadvantage for both users and bike-sharing operators is that it is weather dependent. For this reason, the operation of bike-sharing during the winter is limited. Further disadvantages for the bike-sharing operator are its high start-up costs, including the cost of procuring bicycles, stations and maintenance, and the weak cycling infrastructure in most places when implementing the system. A major disadvantage is also dealing with possible theft and vandalism [9]. There are some other disadvantages of using bike-sharing (e.g., health issues and carrying luggage). People with bad physical fitness, older people and disabled people are not able to use a bike for transportation. Furthermore, there is no space to carry large luggage. In most bike-sharing systems, it is not possible to rent a bicycle adapted to traveling with children. In addition, the necessary accessories are absent, such as a child seat [10].

Most of the published scientific studies are primarily concerned with the composition, operation, analysis and modeling of bike-sharing systems. The available studies can be divided into several different areas of research. First of all, there is empirical research of the functioning and use of a bike-sharing system [11–13].

Furthermore, there are studies on problems with the locations and capacities of bike-sharing stations [14,15].

Additionally, there are other studies aimed at the following:

- Modeling and optimizing problems with the relocation of bicycles [16–19];
- The functioning of the bike-sharing system and the efficiency of using it under different conditions [20];
- The impact of the bike-sharing system on people's health and live as well as on the environment [21–23].

Some studies indicate that the locations of stations being in densely populated areas and places of concentration of companies, schools, universities, restaurants, cinemas and shops increases the number of rentals [24,25].

Until now, studies have been carried out on factors influencing the use of bike-sharing systems and the level of satisfaction with the use of them [26]. The purpose of this research was to understand the factors influencing the relatively low use of the bike-sharing system in Ningbo (China). Based on this research, several conclusions were drawn regarding planning and efficiency to increase the use of this system in the mentioned city.

In the first months of the COVID-19 pandemic, most countries introduced restrictions in the form of reduced mobility to prevent the spread of the virus. Several studies have shown that these restrictions have been able to slow the global spread of this virus [27–29].

The impact of the Covid-19 pandemic has also been intense in the area of urban mobility. In [30], the authors identified a 76% decrease in overall mobility and a 93% decrease in public transport use in Santander, Spain. On the contrary, the decrease in car use was not as significant compared with public transport [30]. In [31], the authors described a significant reduction in traffic of up to 80% in specific countries.

The impact on mobility caused by the COVID-19 pandemic is expected to have a lasting effect on passenger perception and behavior. Many studies [32–34] estimate that this pandemic will lead to a long-term reduction in public transport use and, conversely, an increase in bicycle use and walking. A study carried out in the Netherlands showed that people are now more willing to use cars instead of public transportation [35]. The same study states that a large proportion of people working from home during a pandemic expect to work more often this way in the future [36]. Another study using mobility data from Budapest identified an 80% decrease in demand for public transport and a 2% decrease in demand for bike-sharing [37]. There was a significant decrease in the use of bike-sharing in the case of Beijing, as 40% fewer rentals were made compared with the same period in 2019 [38].

This paper aims to point out the analysis results of the impacts of a bike-sharing system in the city of Košice before and during the COVID-19 pandemic. Last year, there

was a decline in the number of bike rentals due to the impact of the COVID-19 pandemic. In addition, this paper describes an analysis of the current state of bike-sharing in the Slovak Republic. In our research, we asked the following questions:

1. What bike-sharing models are used in Slovakia?
2. Did COVID-19 affect the use of bike-sharing in the selected city?
3. What decrease or increase in the number of bike rentals was recorded in 2020 compared to 2019?

Section 2 describes the bike-sharing system in Slovakia, where we have listed all cities that have established bike-sharing systems. There, the individual bike-sharing systems in the mentioned cities are described and compared. We also describe bike-sharing in Košice in more detail. Košice has the largest system of shared bicycles in Slovakia, which is why we chose this city for our research. In the "Results and Discussion" section, we present a comparison of data for 2019 and 2020. Furthermore, a summary of the results achieved and our conclusions are stated.

## 2. Analysis of the Bike-Sharing System in Slovakia

The following part of our study includes a description of the bike-sharing systems in cities in Slovakia. This includes information on the prices and the number of available bicycles and stations, as well as a comparison of this data between the mentioned cities.

Bike-sharing currently operates in eleven Slovak cities (see Figure 1): Bratislava, Trnava, Piešťany, Nitra, Štúrovo, Prievidza, Žilina, Poprad, Košice, Moldava nad Bodvou and Trebišov [39–46].

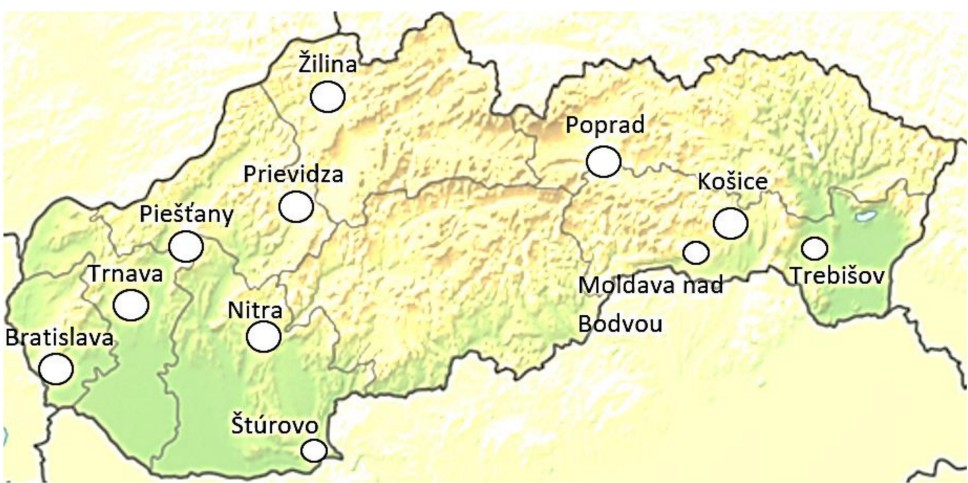

**Figure 1.** Cities in Slovakia providing bike-sharing systems [own study].

*Comparison of Bike-Sharing Systems between the Mentioned Cities*

For comparison between cities, we chose data based on population and city area, among other factors. The population of the city is an indicator of the possible demand for the service operating in the city. The more inhabitants the city has, the more potential users of the service there are. If it is a significant city, there is a growing need for people to commute from home to school or work. This can make bike-sharing more attractive and contribute to its greater use. It is university students who can have a significant impact on the success of bike-sharing. The large area of the city contributes to the need for transport from housing estates and more distant points of the city, which is suitable for the use of bike-sharing.

Table 1 provides data on the population, city area and terrain for each city where a bike-sharing system operates. The bike-sharing model used is also listed.



**Table 1.** Information about the cities where bike-sharing works [own study], based on [39–46].

| City | Model | Population * | City Area (km$^2$) | Terrain |
|---|---|---|---|---|
| Bratislava | Docking station | 437,725 | 367.33 | Rugged terrain |
| Trnava | Geo-fencing | 65,033 | 71.54 | Flat terrain |
| Nitra | Docking station | 76,533 | 100.48 | Rugged terrain |
| Piešťany | Free-floating | 27,336 | 44.2 | Flat terrain |
| Prievidza | Docking station | 50,626 | 63.02 | Flat terrain |
| Žilina | Docking station | 80,727 | 80.03 | Rugged terrain |
| Košice | Free-floating | 238,593 | 242.77 | Rugged terrain |
| Poprad | Free-floating | 51,235 | 63.11 | Rugged terrain |
| Trebišov | Free-floating | 24,649 | 70.16 | Flat terrain |
| Moldava nad Bodvou | Free-floating | 11,307 | 19.77 | Rugged terrain |
| Štúrovo-Ostrihom | Docking station | 39,330 | 114.4 | Flat terrain |

* For the period of 2020 (in thousands).

If the terrain is difficult for the user to overcome with a bicycle, this may discourage him or her from choosing this means of transport. Some bike-sharing providers, such as Arboria bike in Trnava, use electric bicycles for transportation [43]. Antik Telecom s.r.o. also plans to put into operation electric bicycles. This company also operates electric scooters in Košice, which allow easier access to hilly terrain and are also suitable for longer routes [39].

The use of this system is also affected by the number of available bicycles for transportation. There must be no situations where the user wants to rent a bike, but there is no free bike available at the moment, or it is too far away. This could affect the satisfaction with the service and its further use.

Table 2 shows the information on the individual systems operating in the cities. One can see the number of available bicycles and stations as well as the number of inhabitants per bicycle available in the program, the number of bicycles calculated per 1000 inhabitants and the number of stations per 1 km$^2$.

**Table 2.** Information on the number of bicycles and bike-sharing system locations in individual cities [own study], based on [39–46].

| City | Number of Bicycles | Number of Stations | Population/1 Bicycle | Number of Bicycles/1000 Inhabitants | Number of Stations/km$^2$ |
|---|---|---|---|---|---|
| Bratislava | 750 | 96 | 584 | 1.71 | 0.26 |
| Trnava | 118 | 90 | 551 | 1.81 | 1.26 |
| Nitra | 70 | 7 | 1093 | 0.91 | 0.07 |
| Piešťany | 120 | 6 | 228 | 4.39 | 0.14 |
| Prievidza | 103 | 25 | 492 | 2.03 | 0.40 |
| Žilina | 120 | 20 | 673 | 1.49 | 0.25 |
| Košice | 1000 | 91 | 239 | 4.19 | 0.37 |
| Poprad | 65 | 12 | 788 | 1.27 | 0.19 |
| Trebišov | 65 | 12 | 379 | 2.64 | 0.17 |
| Moldava nad Bodvou | 25 | 6 | 452 | 2.21 | 0.30 |
| Štúrovo-Ostrihom | 142 | 20 | 270 | 3.70 | 0.17 |

The number of inhabitants per bicycle shows us the possible demand for the bicycle in the bike-sharing system. The last column of the table describes the density of stations in individual cities. However, this is not exact information, as the stations are deployed mainly according to user needs. Nevertheless, we can see in which cities there are fewer stations concerning the city area. The fewest number of stations per km² is in Nitra at 0.07 stations per km². On the contrary, the best result is in Trnava (1.26 stations per km²), but this is also caused by using a different bike-sharing model. The stations in Trnava are only virtual, and there is no need to build physical stations. The largest systems in Slovakia are in SlovnaftBajk in Bratislava and Verejný bicykel in Kosice [39–46]. Figure 2 shows the ratio of the number of bicycles in the individual bike-sharing systems.

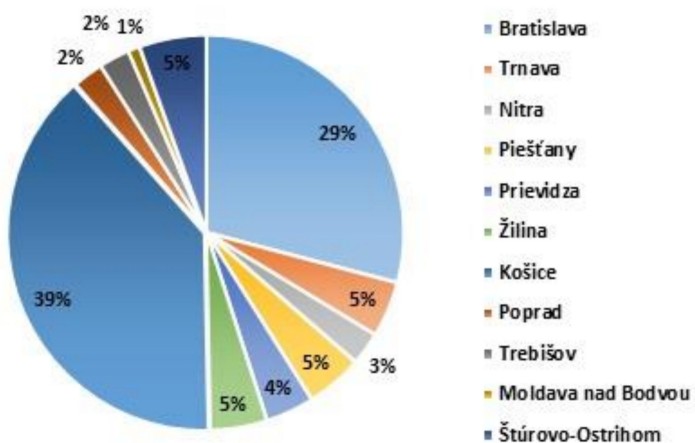

**Figure 2.** The ratio of the number of bicycles in the bike-sharing systems in cities in Slovakia [own study], based on [39–46].

We can see that the largest number of shared bicycles is in Košice, and the second largest number is in Bratislava [39,41]. These two cities are the largest cities in the Slovak Republic, and there are also the largest numbers of bicycles.

Table 3 describes the number of bicycle rentals and the number of users per bicycle available in the provider's system. The data are for the year 2019. The highest value was in Nitra and Žilina. In Nitra, this was due to the smaller number of bicycles available in the system.

**Table 3.** Information on the number of users and bicycle rentals in cities where bike-sharing systems are operated (2019) [own study], based on [39–46].

| City | Population | Number of Users * | Number of Bicycle Rentals * | Number of Users/1 Bicycle |
|---|---|---|---|---|
| Bratislava | 437,725 | 44,000 | 440,000 | 59 |
| Trnava | 65,033 | 1800 | 110,000 | 15 |
| Nitra | 76,533 | 14,000 | 45,000 | 200 |
| Piešťany | 27,336 | Unknown | Unknown | Unknown |
| Prievidza | 50,626 | 4407 | 13,632 | 43 |
| Žilina | 80,727 | 23,000 | 291,000 | 192 |
| Košice | 238,593 | | 226,727 | |
| Poprad | 51,235 | 24,000 together | 6100 | 14 |
| Trebišov | 24,649 | | 3886 | |
| Moldava nad Bodvou | 11,307 | | 1249 | |
| Štúrovo-Ostrihom | 38,330 | Unknown | Unknown | Unknown |

* Approximate value [39–46].

Data from the system in Piešťany were not available, as it was a system without registration; therefore, data on the number of users were unavailable. Additionally, we had no data from Štúrovo-Ostrihom.

A comparison in terms of the price for renting a bike is one of the important factors in determining the attractiveness of the system. Some providers offer free minutes at the start of the ride, which can attract users. If the customer can get to the designated place by bicycle and knows that he or she will be able to do so within the free limit, he or she is more likely to choose this service.

For comparison, we chose to rent a bicycle for 30 min, one hour and two hours. This can be seen in Table 4 [39–46], which shows the prices in individual cities when renting a bicycle for 30 min, 1 h and 2 h.

**Table 4.** Prices for bicycle rentals in individual cities with rental periods of 30 min, 1 h and 2 h [own study], based on [39–46].

| City | The Price of Driving at a Given Time (EUR) | | |
|---|---|---|---|
| | 30 min | 1 h | 2 h |
| Bratislava | 0.60 | 0.60 | 1.80 |
| Trnava | 0.35 | 1.00 | 2.50 |
| Nitra | 0.50 | 0.50 | 1.00 |
| Piešťany | 0.0 | 0.0 | 0.0 |
| Prievidza | 0.40 | 0.80 | 1.20 |
| Žilina | 0.0 | 0.0 | 0.0 * |
| Košice | 1.00 | 1.00 | 2.00 |
| Poprad | 1.00 | 1.00 | 2.00 |
| Trebišov | 1.00 | 1.00 | 2.00 |
| Moldava nad Bodvou | 1.00 | 1.00 | 2.00 |
| Štúrovo-Ostrihom | 1.30 | 1.80 | 2.80 |

\* Free of charge for 1 h.

Figure 3 shows a comparison of the prices in individual cities. The blue color shows the rental time of the bike for 30 min, the orange is for 1 h, and the gray is for 2 h.

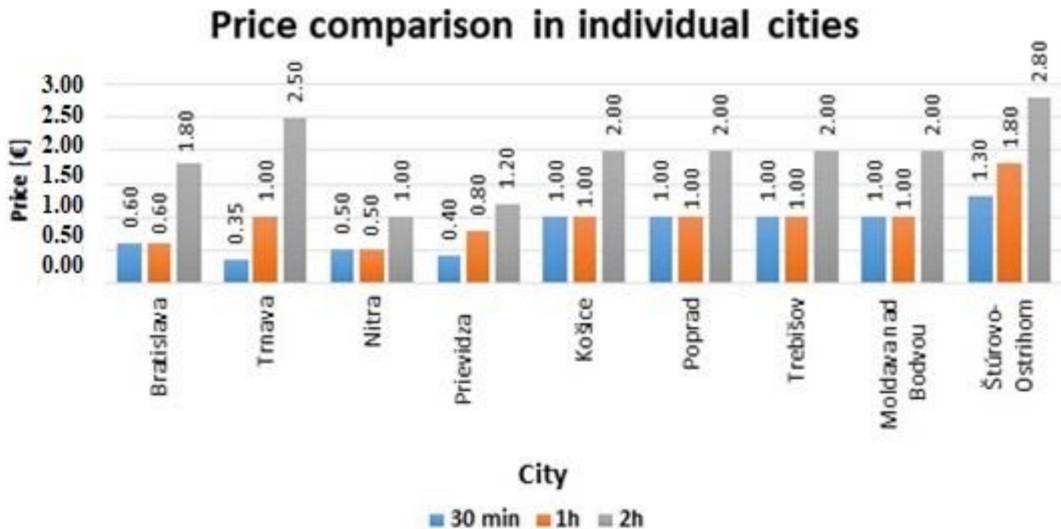

**Figure 3.** Price comparison when paying for bike rides with times of 30 min, 1 h and 2 h [own study], based on [39–46].

In Bratislava, we chose a credit program for comparison, with a principle of deduction of credit for the minutes of rental [41]. There is no possibility of choosing programs or packages in Trnava. The amount to be paid is proportional to the rental time [43]. In Nitra, we chose a one-off fee, in which the rental price is cumulated with the rental time [40]. "Piešťanské užitočné koleso" in Piešťany is a free service [45]. In Prievidza, we chose a one-time payment via SMS [44]. In Žilina, the first hour of the rental is free, and then we can use the opportunity to return the bike and rent it again for another free hour. If we did not lock the bike, we would be charged EUR 20 for a ride over the limit of 1 h [42]. There is a shared tariff for Košice, Poprad, Trebišov and Moldava nad Bodvou [39]. In terms of price, the most favorable systems are "Piešťanské užitočné koleso" and "BikeKIA" in Žilina.

The accessibility of the individual systems implemented in the mentioned cities differs not only in price, but also in limiting the minimum age of the user. Table 5 shows the minimum age of the user in each city.

**Table 5.** Minimum ages of bike-sharing system users in individual cities [own study], based on [39–46].

| City | Minimum User Age |
|---|---|
| Bratislava | 16 |
| Trnava | 18 |
| Nitra | 16 |
| Piešťany | Unspecified |
| Prievidza | Unspecified |
| Žilina | 16 |
| Košice | 18 |
| Poprad | 18 |
| Trebišov | 18 |
| Moldava nad Bodvou | 18 |
| Štúrovo-Ostrihom | 14 |

Table 5 shows the compared advantages and disadvantages of the individual bike-sharing systems in Slovakia. The table is based on the described analysis of the bike-sharing systems in Slovakia.

Table 6 shows that in terms of price, the most advantageous bike-sharing is in Piešťany. The system in Piešťany is the easiest to use for users. On the one hand, it is free. On the other hand, there is no need to install any application, and the bike is always available for immediate use without the need to unlock or lock the bike. This system would be more attractive with more bicycles to rent. Therefore, we consider bike-sharing in Žilina to be the most advantageous. It offers the possibility to return the bike and rent it again, while the user gets a free rental hour. This offers users plenty of time to travel for free and provides more bicycles than the bike-sharing system in Piešťany.

**Table 6.** Advantages and disadvantages of individual bike-sharing systems in Slovakia [own study], based on [39–46].

| SlovnaftBajk—Bratislava | | Arriva bike—Nitra | |
|---|---|---|---|
| *Advantages* | *Disadvantages* | *Advantages* | *Disadvantages* |
| A large number of bicycles | Deposit EUR 30 for the daily program | Possibility to use nextbike programs | Small number of bicycles |
| Does not require installing the application | | | Small number of bicycle stations |
| **Arboria bike—Trnava** | | **Piešťanské užitočné koleso—Piešťany** | |
| *Advantages* | *Disadvantages* | *Advantages* | *Disadvantages* |
| Virtual stations | When renting for 2 h, one of the most expensive programs | Free use | All bicycles are often rented |
| Electric bicycles | | Without the need to lock | |
| Possibility of pause | | | |
| **Zelený Bicykel—Prievidza** | | **Verejný bicykel—Košice, Trebišov, Poprad, Moldava nad Bodvou** | |

### 3. Bike-Sharing System in Košice before and during the COVID-19 Pandemic

We chose the city of Košice because it has the largest bike-sharing system in the Slovak Republic. Also, it is the second-largest city in the Slovak Republic.

Košice is one of the six cities in Slovakia where there is a bike-sharing system from Antik Telecom s.r.o. The system operator was willing to provide us with data. The full operation was launched in May 2019. The use of rented bicycles is not limited to the city area, but their return must be made in this area. The operation of this system is based on the free-floating model [39].

Stations have been set up in the city where it is possible to park and lock a bicycle [39]. However, this is not a condition, and the user can lock the bicycle freely in the city, provided that the safety and smoothness of traffic is maintained. One cannot park the bicycle on a sidewalk, road or in a place where it would restrict the movement of pedestrians and cyclists. All bicycles are equipped with a mechanical system on the rear wheel which allows them to be locked away from the station.

As for bike-sharing in Košice, it has the largest number of bicycles in Slovakia. Users have at their disposal approximately 1000 smart bicycles, each equipped with a GPS. There are 91 stations. It is necessary to register to use this service. The applicant can do this using the Antik SmartWay mobile application [39,47].

After registration, it is necessary to pay a deposit of EUR 20. This deposit is returned to the user's bank account in the case of the cancellation of the user's account. It is also necessary to have at least EUR 1 of credit [39].

The prices for bicycle rental are as follows: payment after the rental is EUR 1; the monthly payment is EUR 5; and the seasonal payment is EUR 30. Each hour, a fee of EUR 1 is charged. The characteristics of the bike-sharing system are in Table 7.

**Table 7.** Characteristics of the bike-sharing system [own study], based on [39,48].

| City | Population | Area (km²) | Start of Operation | Operator | Service Name | Model | Number of Stations | Number of Bicycles |
|------|-----------|-----------|-------------------|----------|-------------|-------|-------------------|-------------------|
| **Košice** | 238,593 | 242.77 | May 2019 | Antik Telecom s.r.o. | Verejný bicykel | Free-floating | 91 | 1000 |

#### 3.1. COVID-19 in the Slovak Republic

The first case of COVID-19 (in Slovakia) was confirmed on 6 March 2020. Extensive measures against the spread of COVID-19 were gradually taken. Schools, offices, churches and some factories and shops (except groceries) were closed. These strict measures led to a reduction in people's mobility. Additionally, they significantly reduced the spread of COVID-19. During June, the number of people with COVID-19 increased minimally. Gradually, individual measures were relieved. This favourable situation also resulted in increased mobility.

The situation deteriorated in September and led to a tightening of measures. Based on that, the government of the Slovak Republic declared a second state of emergency (1 October 2020). Exemptions from the curfew changed after mass testing. Fitness centers, swimming pools, churches and some schools, among other places, were gradually opened under strict hygiene rules. The situation started to deteriorate again after the relieving of measures. At the end of the year, there was a strict lockdown (churches, schools and other places were closed).

Figure 4 is a timeline showing significant dates during 2020 related to the COVID-19 pandemic.

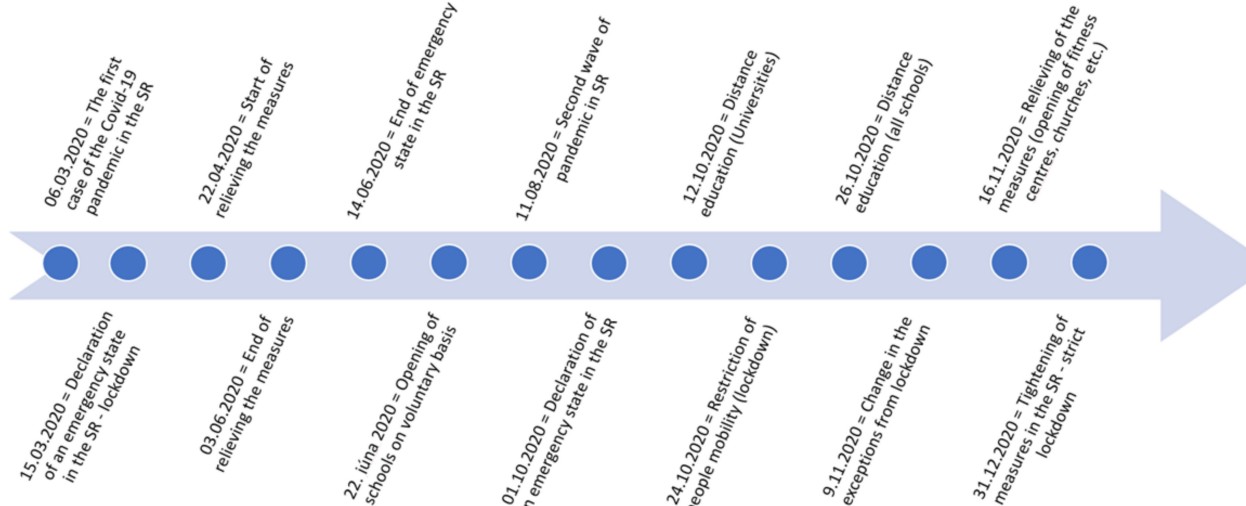

**Figure 4.** Timeline of significant dates during 2020 related to the COVID-19 pandemic [49–52].

Figure 5 shows the course of the number of people with COVID-19 from March to December 2020.

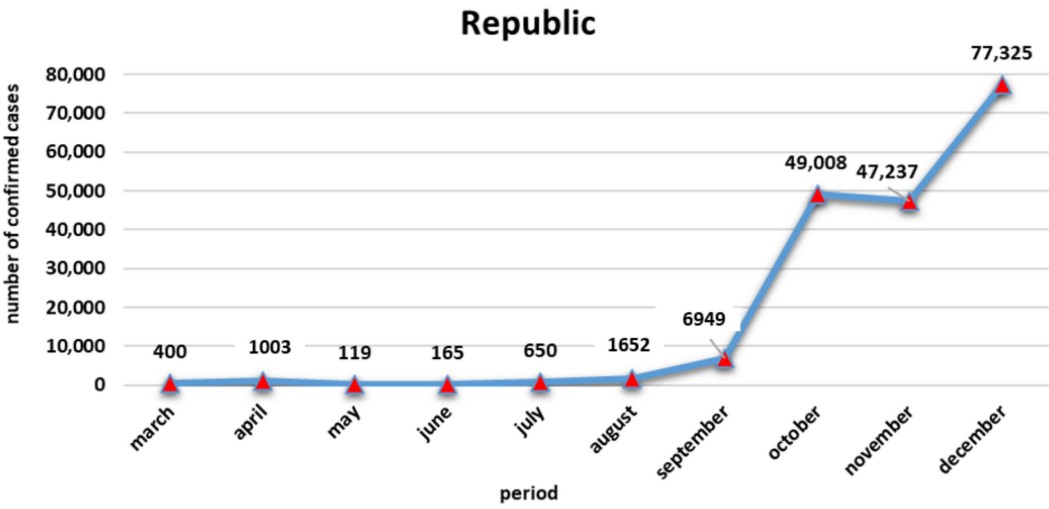

**Figure 5.** The course of the number of people with COVID-19 in the Slovak Republic [51].

### 3.2. Data Processing Research Methodology

This part of the paper aims to create a uniform and comprehensive procedure for the data analysis so that this procedure can be used in the same way and enable monitoring of the development of bike-sharing use in individual cities.

This procedure was divided into three basic parts:

- Data pre-processing;
- Non-spatial analysis;
- Spatial analysis of data processing.

Together, these parts formed a complete guide on how to process data from the export from the database to the creation of the presentable outputs (e.g., graphs, tables and map outputs).

Methodology and Processing Procedure

For the purposes of this paper, spatial and non-spatial analysis were used. Spatial analysis is the process of examining trends, equations and relationships between elements and phenomena in a territory. Spatial analysis extracts or creates new information from the input spatial data. It is used by geographic information systems [53].

An important part of data processing was data pre-processing, followed by spatial and non-spatial analysis. Together, these parts formed a comprehensive procedure for analyzing the processing of bike-sharing data.

- **Used data:**

The most important data collected was information on the use of the Antik Telecom s.r.o. bike-sharing system. Specifically, these were data for the year 2019, when bike-sharing was introduced in Košice, and data for the year 2020, during which the COVID-19 pandemic situation began and persisted. These data were for the period from May to December.

Data from Antik Telecom s.r.o. were provided in .csv text format. A total of 237,962 records for 2019 and 131,016 records for 2020 were processed (e.g., date and time of bicycle rental, GPS route coordinates and bicycle identification data).

- **Used programs:**

Microsoft Excel was used to process the non-spatial analysis. The Quantum Geographic Information System (QGIS) desktop traffic planning program version 3.4.15 [54] and Open Street Maps [55] were used to process the spatial accessibility analysis.

- **Processing procedure:**

The following procedure was chosen for the processing of the provided data:

○ The study of the scientific literature dealing with the analysis of data on bike-sharing;
○ Obtaining the provided data (2019 and 2020) from Antik Telecom s.r.o.;
○ Pre-processing of the data provided by the operators into the form in which they would enter the analysis;
○ Processing of non-spatial analysis and the creation of graphs and tables with monthly, weekly and daily reports;
○ Processing of spatial analysis and the creation of maps on the use of bike-sharing;
○ Completion of information in the literature review and finishing the text part of the paper.

*3.3. Pre-Processing of Data*

Data pre-processing was a very important phase which meant the removal of all unsuitable data that was provided. The resulting data was then prepared for non-spatial or spatial analysis. All results and outputs from these analyses were affected by the quality of the data pre-processing, which made this phase very important.

Some of the records in the data set did not occur in the territory of Košice, and in some instances, part of the data gathered contained records from another city (e.g., Poprad and Trebišov).

This was because the identifier of a bicycle in the database remained the same even after relocation to another city. For this reason, all bicycle activity records were transferred from all cities where the bicycle was in operation. Some records did not even contain any spatial localization, and therefore, they were not usable for further processing [39].

In some cases of data pre-processing, there was a small decrease in the number of records for each data set. However, this decrease occurred because some of the bicycles were involved in testing the system during the 2019 season, so we did not use this data for data processing. A total of 365,949 records were processed (2019 = 237,962 records; 2020 = 127,987 records).

For these reasons, it was important to carry out pre-processing of the data correctly, which guaranteed that the correct records of the rentals would enter the analyses.

## 4. Results and Discussion

This section describes the statistical evaluation of the data on the use of bike-sharing in Košice. A comparison between 2019 and 2020 is also described. Part of our research was for examining the impact of the COVID-19 pandemic on the use of bike-sharing.

The COVID-19 pandemic and individual measures had a significant impact on people's mobility. Figure 6 shows the course of people's mobility at public transport stations in the Kosice region.

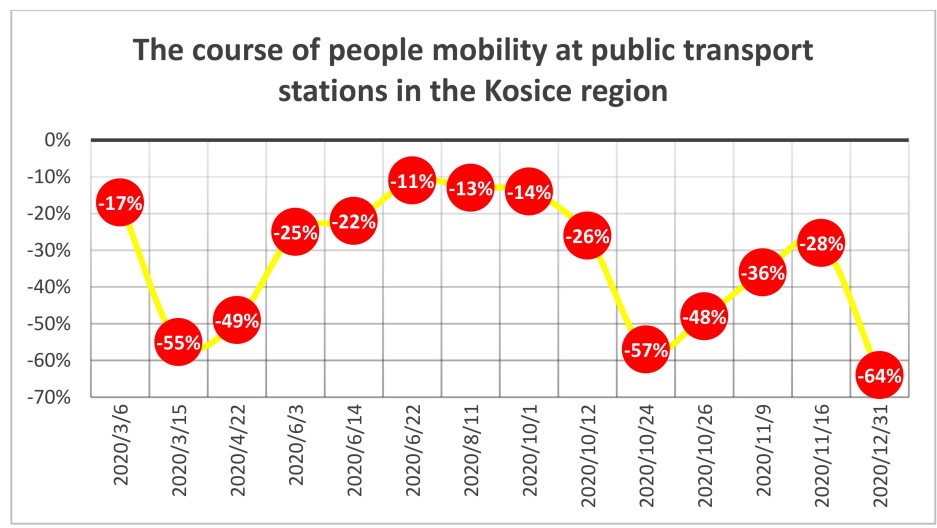

**Figure 6.** The course of people's mobility at public transport stations in the Kosice region [56].

A comparison of the decline in the use of bike-sharing and the reduction in people's mobility at public transport stations is in Figure 7. We performed this comparison based on the individual important dates shown in Figure 4. However, bike-sharing data for March–May were not available. Therefore, the values were missing for these months (6 March, 15 March and 22 April 2020). The course of reduced mobility at public transport stations was a reflection of the measures at the time. The comparison of bike-sharing use for the mentioned dates revealed an interesting increase in November and December. The reason for this increase is stated in the text below Figure 8.

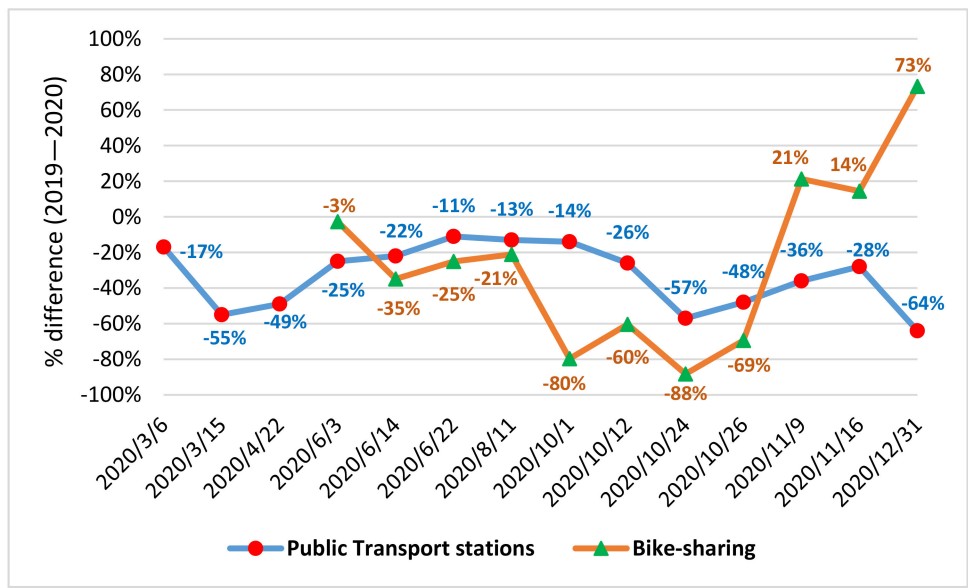

**Figure 7.** The decline in the use of bike-sharing and reduction in people's mobility at public transport stations [authors].

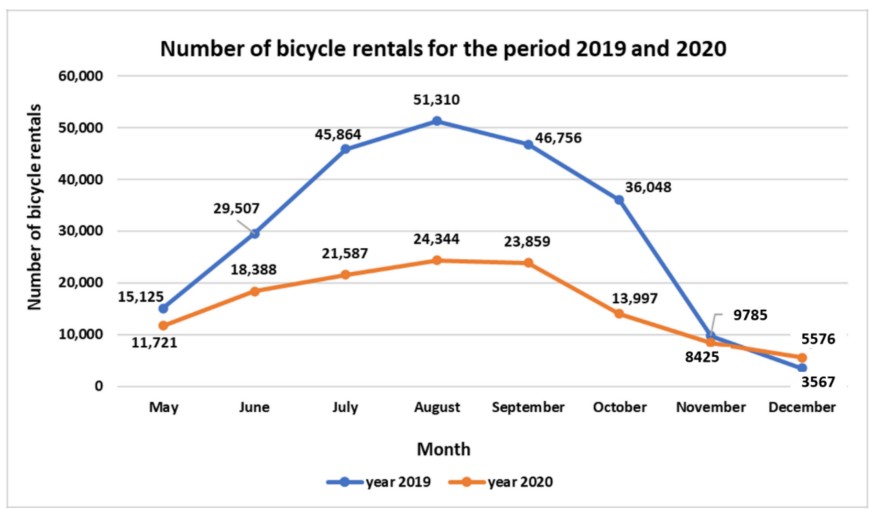

**Figure 8.** Number of bicycle rentals for the periods of 2019 and 2020 [own study], based on [39].

In addition, in [52], the authors described the results of how the COVID-19 pandemic affected the mobility of the Slovak population. The authors stated that the decrease in mobility was due to the impact of COVID-19. The most significant decrease was recorded for the following passenger groups: students, children (under 6 years old) and seniors (65 years and older).

Table 8 shows the number of bicycles rented from May to December in 2019 and 2020. The table shows that in 2020, there was a 46.25% decrease in bicycle rentals compared with 2019 (a decrease of 110,065 rentals). This decrease was caused by COVID-19. It had a very significant impact, as evidenced by major changes in the decrease in bicycle rentals.

**Table 8.** Number of bicycle rentals via Antik Telecom s.r.o. for the years 2019 and 2020 [own study], based on [39].

| Month | | May | June | July | August | September | October | November | December | Together |
|---|---|---|---|---|---|---|---|---|---|---|
| Number of Bicycle Rentals | year 2020 | 11,721 | 18,388 | 21,587 | 24,344 | 23,859 | 13,997 | 8425 | 5576 | 127,897 |
| | year 2019 | 15,125 | 29,507 | 45,864 | 51,310 | 46,756 | 36,048 | 9785 | 3567 | 237,962 |
| Difference (2020–2019) | | −3404 | −11,119 | −24,277 | −26,966 | −22,897 | −22,051 | −1360 | 2009 | −110,065 |
| % Decrease or Increase | | −22.51% | −37.68% | −52.93% | −52.56% | −48.97% | −61.17% | −13.90% | 56.32% | −46.25% |

Figure 8 shows the course of bicycle rentals from May to December in 2019 and 2020. The figure also shows a significant decrease of 52.56% in bicycle rentals in August 2020 compared with August 2019. From September to December, a decrease in the number of bicycle rentals can be observed (due to worse weather conditions (i.e., cold, rainy weather)). The increase in the number of bicycle rentals in December 2020 was due to the favorable weather during this period and the relieving of restrictions (e.g., opening shops, services, churches and schools) against the spread of COVID-19. According to data from the source [57], December was strongly above normal (very warm) in most parts of Slovakia, except in some locations. Deviations of the average daily air temperature, compared with the normal value for 1981–2010, reached from +5 to +10 °C.

We performed a statistical analysis where we calculated the mean, mode, median and standard deviation. The mode was the most common value in the statistics file. The mean is the usual average and can be calculated according to the following equation [58]:

$$\bar{x} = \frac{1}{n} \sum_{i=1}^{n} x_i \tag{1}$$

where $x_i$ is the value of the *i*th point in the data set and *n* is the number of data points in the data set.

The median is the middle value of the measured values, ordered in a non-decreasing sequence. For an even *n* value [58], the median is calculated as follows:

$$\widetilde{x} = \frac{x_{\left(\frac{n}{2}\right)} + x_{\left(\frac{n}{2}+1\right)}}{2} \tag{2}$$

The standard deviation is a statistic that measures the dispersion of a dataset relative to its mean [59]:

$$\sigma = \sqrt{\frac{\sum(x - \bar{x})^2}{n}} \tag{3}$$

In statistics, the variance measures the variability from the average or mean. The variance is calculated using this equation [59]:

$$\sigma^2 = \frac{1}{n} \sum_{i=1}^{n} (x_i - \bar{x})^2 \tag{4}$$

The results of the statistical analysis are stated in Table 9. It is possible to see an increase in the rental time in 2020 compared with 2019. This increase is represented by both the mean (11.03%) and the median (14.56%).

**Table 9.** Results of statistical analysis of the average bicycle rental time [authors].

| Period | Mean | Median | Standard Deviation | Variance |
|--------|------|--------|--------------------|----------|
| 2020 | 9.96 | 10.15 | 1.4 | 1.69 |
| 2019 | 8.97 | 8.86 | 1.15 | 1.32 |

Figure 9 shows the course of the average bicycle rental time from May to December for both years (2019 and 2020).

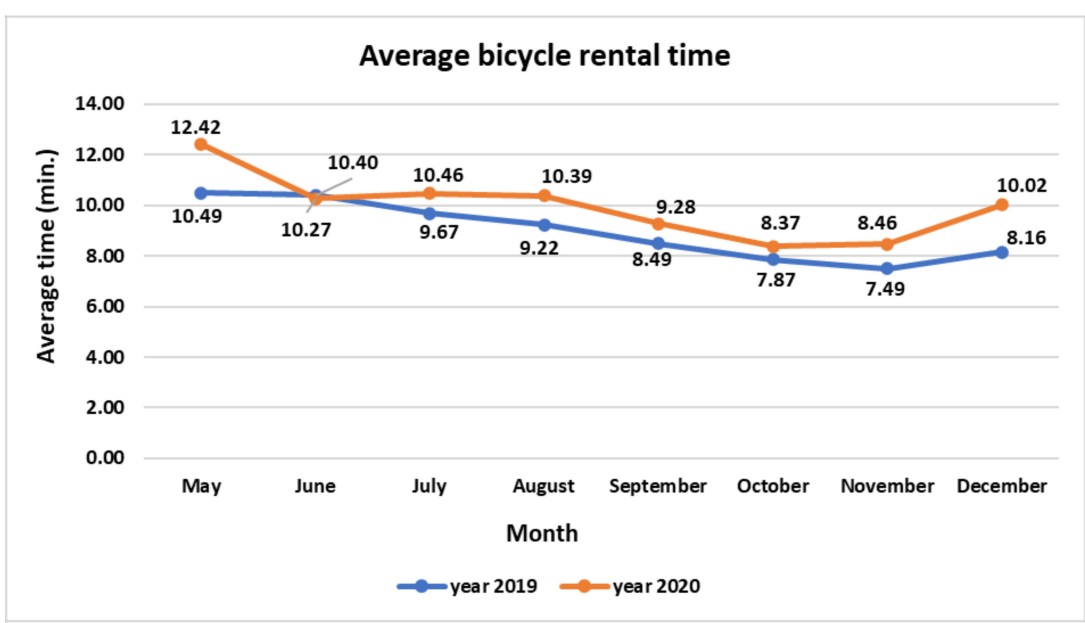

**Figure 9.** Average bicycle rental time for the periods of 2019 and 2020 [own study], based on [39].

Figure 10 shows rentals of bicycles for each day of the week (2019 and 2020). The days with the largest number of rentals included Monday, Tuesday, Wednesday, Thursday and Friday. These days represent workdays, when the traffic volume is higher than during weekends.

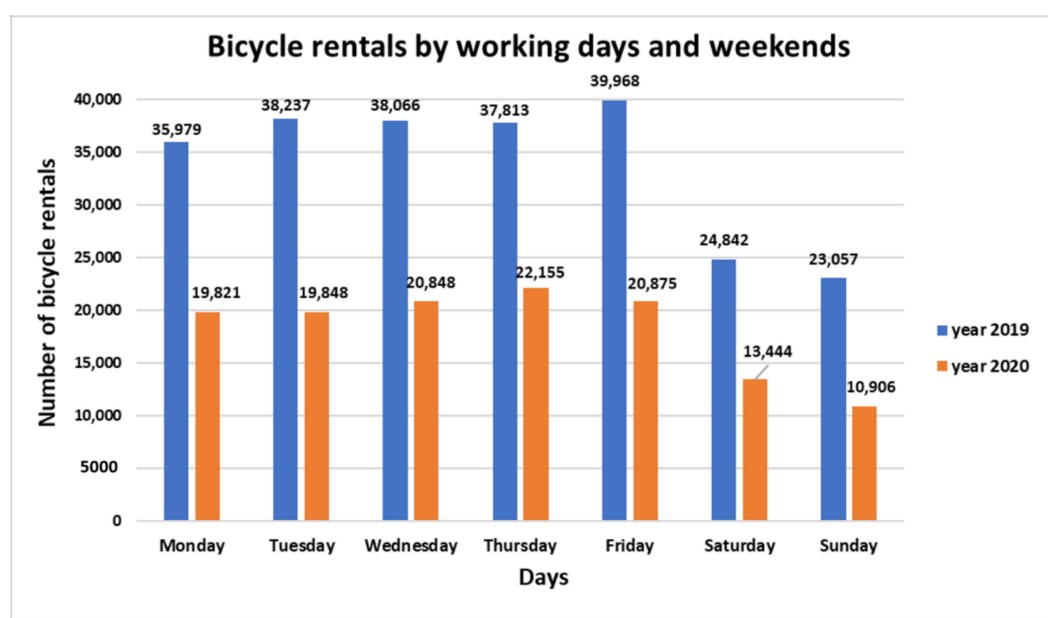

**Figure 10.** Bicycle rentals by working days and weekends for the periods of 2019 and 2020 [own study], based on [39].

Figure 11 shows the course of bicycle rentals during the day in 2019 and 2020. In 2019, the maximum number of rentals was in the interval from 4:00p.m. to 5:00 p.m. (19,620 rentals). Additionally, in 2020, the maximum (11,119 rentals) was recorded in the same interval. This represents the afternoon peak hour. During this time interval, many people are going home from work, schools and shops. During evening and night hours, the number of rentals was lower than during the day, when people's mobility was higher.

The morning peak hour was between 7:00 a.m. and 08:00 a.m. The number of rentals for each year was 11,230 (2019) and 5792 (2020).

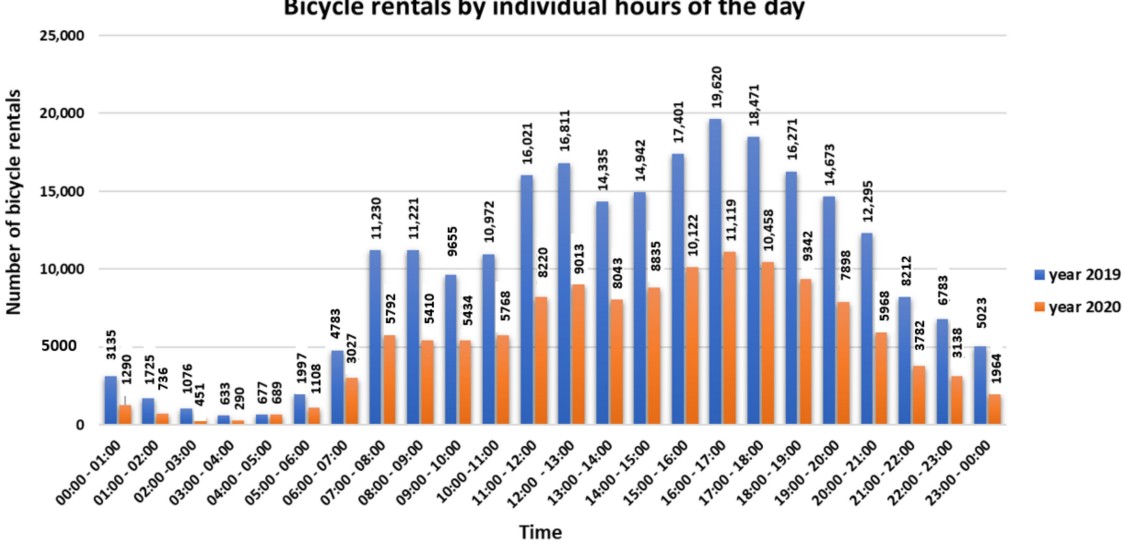

**Figure 11.** Bicycle rentals by individual hours of the day for the periods of 2019 and 2020 [own study], based on [39].

There was a significant decrease in bicycle rentals in Košice during the COVID-19 pandemic. The bike-sharing system was not used often for transportation to school, work or other locations because most people worked and studied from home.

*Data Comparison: August 2019 and 2020*

This part describes the comparison of data for August (2019 and 2020). We chose this month due to the maximum number of bicycle rentals during both compared years. Additionally, the most significant decrease in bike rentals was in 2020, compared with 2019.

Figure 12 shows the number of rentals for each day of the week. The days with the largest number of rentals in 2019 included Thursday and Friday. The maximum was recorded on Friday: 9656 rentals. In 2020, there were 64.5% (6228) fewer bicycle rentals on Friday.

**Figure 12.** Bicycle rentals by working days and weekends for August 2019 and 2020 [own study], based on [39].

Figure 13 shows the course of bicycle rentals at hourly intervals throughout the day. The course of bicycle rentals was the same during both years, but in 2020, there was a significant decrease in the number of rentals.

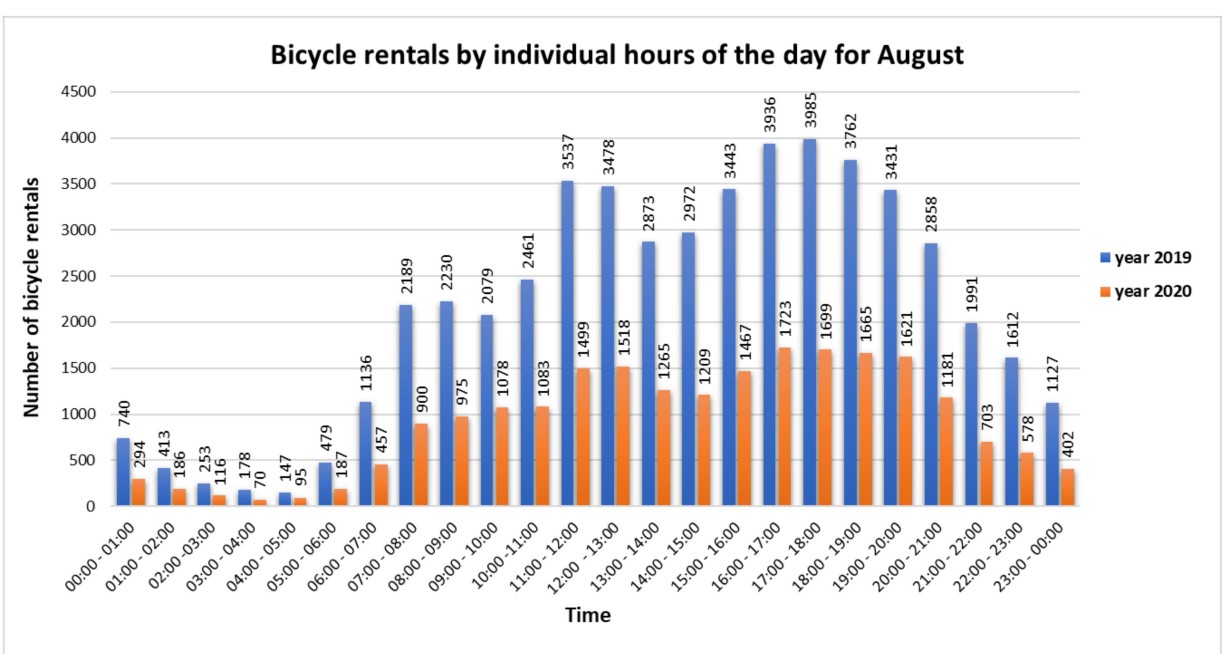

**Figure 13.** Bicycle rentals by individual hours of the day for August 2019 and 2020 [own study], based on [39].

The main purpose of the spatial analysis was to analyze the data provided on bike-sharing from Antik Telecom s.r.o. We performed this analysis using the selected GIS software. Figure 14 shows that there was a larger volume of transportation in 2019 (Figure 14a) compared with 2020 (Figure 14b). The largest changes can be seen in the outskirts of the city. One can see a higher density of routes in 2019, such as in the western or northeast parts of the city. In 2019, there were 51,310 rentals in Košice, while in 2020, there were only 24,344 rentals.

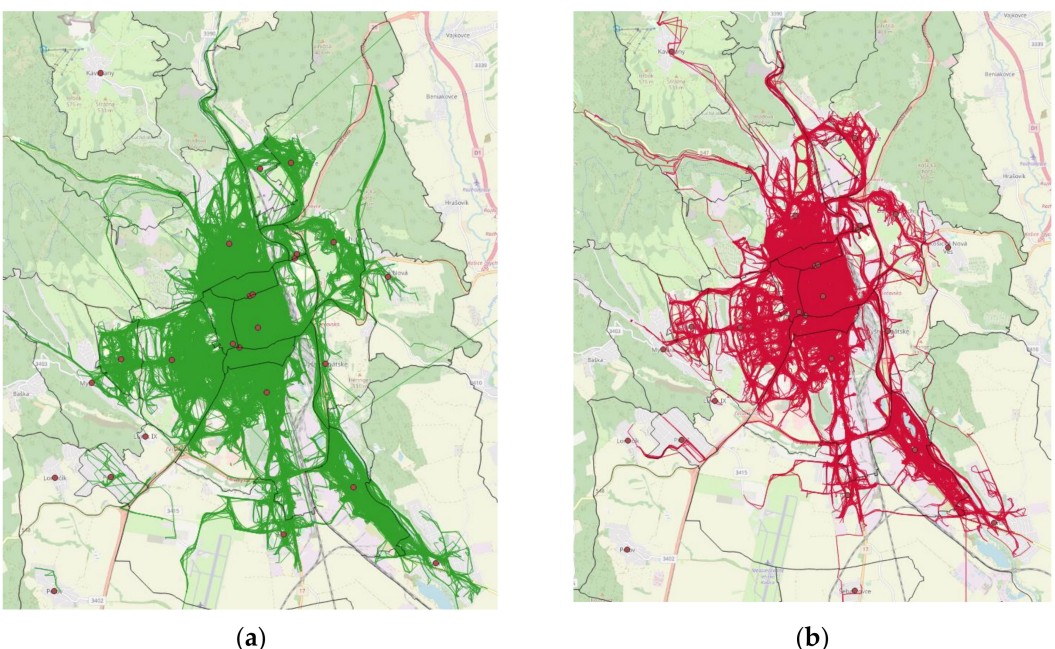

(**a**)  (**b**)

**Figure 14.** (**a**) Tracing of rented bicycles in August 2019 and (**b**) August 2020 [own study].

In our research, we found answers to our research questions. In Slovakia, there are three types of bike-sharing models in use: docking station, geo-fencing and free-floating. The most-used models are the free-floating and docking station models.

The COVID-19 pandemic affected bike-sharing in Slovakia. In addition, individual measures against the spread of COVID-19 influenced mobility and bike-sharing in Slovakia. It is possible to see that impact in the graphs, tables and results described in the third and fourth sections of the paper.

In 2020, there was a 46.25% decrease in the number of bike rentals (a decrease of 110,065 rentals). A decrease was achieved throughout the year, except in December, when an increase of 56.32% was recorded.

The largest decrease was in October (61.17%). The main reason for this decrease was the lockdown during this month. Restrictions were relieved in November, and demand for transport increased slightly. There was a smaller percentage of decrease between 2019 and 2020.

The decrease was caused by COVID-19, which had a very significant impact on the bike-sharing system. As we mentioned, the system was not used because most people worked and studied from home (the home office). However, it was not only caused by the home office and other restrictions. It was also caused by the people's fear of COVID-19. Many people were afraid of using the bike-sharing system or public transport, where they would meet other passengers. People's fear was mainly caused by the risk of infection. Bicycles are not disinfected after each ride, as it is very difficult to ensure disinfection. In this way, it is possible to get infected if the bike was used by someone with COVID-19. It is simply caused by touching the bike with hands.The user had to wear a face mask and gloves when renting a shared bicycle [39]; nevertheless, there were people without a face mask or gloves. For some, it was uncomfortable to wear it while they were using a shared bicycle. People preferred to use their own car or bicycle for transportation.

However, interesting results were achieved for the average rental time of the bicycles. In 2020, there was a longer average rental time compared with 2019 during all months except June. The average rental time in 2019 was 8.97 min, and in 2020 it was 9.96 min. Thus, in 2020, the average rental time was almost 1 min longer. This is an increase of approximately 11%. A longer rental time is a very interesting result. We do not have a rational justification for this result. On the one hand, there was restricted movement for people (lockdown). On the other hand, the rental time increased, but the number of rentals decreased. It is possible that people started using bicycles for longer routes and enjoyed cycling more. Additionally, people lacked social contact and enjoyed talking to other people in real life (not just via video call). We think that before the pandemic, there was a more hectic life in Slovakia. During the pandemic, people began to enjoy the opportunity to meet another person. This could also have caused longer rental times. Of course, we cannot say that this assumption of ours is the real reason for longer rental times. However, this is one of the options.

In Košice, the flow of bicycles is higher in the city center area. On the other hand, bicycles are located throughout the city. The purpose of bike-sharing is to offer an alternative, more environmentally friendly mode of transport.

The decrease in using bike-sharing systems will influence the demand for transport and its sustainability. The bike-sharing companies will have a little loss of income, but all companies provide at least half an hour of use for free. Most rentals were provided for free before this pandemic, so we can argue that it will have no significant impact on the income of providers. However, due to the COVID-19 pandemic, the provider Antik halved the monthly subscription (from EUR 5 to EUR 2.5). Antik wants to attract people to start using bike-sharing again this way. We still believe that after this pandemic, people will use shared bicycles again and more often. Public policy should support people using bike-sharing for transportation, as it could help to attract people.

System operators are required to have enough bicycles at a given time and place that can meet demand and enable the short-term use of this sustainable and environmentally

friendly mode of transport. Bicycles can be rented and returned at freely accessible stations located in designated places around the city. Today, these stations allow smooth operation without the need for an operator thanks to full automation.

The introduction of the bike-sharing system for residents and visitors to cities has great potential. This is also evidenced by the increase in users of bike-sharing around the world.

There are several positive benefits to sharing public bikes:

- An affordable alternative to individual car transport;
- Reduction of the amount of greenhouse gas emissions produced;
- Increasing the mobility of residents and visitors to the city;
- Increasing the attractiveness of the city from the point of view of tourism.

### 5. Conclusions

Our paper aimed to examine the impact of the COVID-19 pandemic on the bike-sharing system. For our research, we used data from 2019, when the bike-sharing system was not affected by the COVID-19 pandemic. We compared these data with data from 2020, when COVID-19 significantly affected this system. In addition, our other goal was to describe the functioning of the bike-sharing system in cities in Slovakia. In recent years, these systems have been introduced in many cities in Slovakia. Bike-sharing represents a method of sustainable and smart mobility.

The quality of the bike-sharing system should meet the user's expectations. Therefore, it is possible to fulfill a questionnaire where users can state any expectations, requirements and possible solutions to improve bike-sharing attractiveness. The achieved results may be useful in carrying out activities aimed at increasing the use of bike-sharing as well as increasing the user's level of satisfaction.

In our further research, we want to focus on the outputs of this questionnaire. We also want to find out the behavior and age structure of the users. It will be interesting to continuously monitor the development of the number of shared bicycle rentals during the COVID-19 pandemic and subsequently when it ends. Will the demand for bike-sharing be the same as before the COVID-19 pandemic, or will it decrease or increase?

**Author Contributions:** Conceptualization, S.K. and A.H.; data curation, S.K., A.K. and A.H.; formal analysis, S.K. and A.H.; investigation, S.K.; methodology, S.K.; resources, S.K.; visualization, S.K., A.K. and A.H.; writing—original draft, S.K. and A.H.; writing—review and editing, S.K.; supervision, A.K.; project administration, A.K.; funding acquisition, A.K. All authors have read and agreed to the published version of the manuscript.

**Funding:** This paper was created within the Identification and possibility of implementing new technological measures in transport to achieve safe mobility during the pandemic caused by COVID-19 (ITMS 313011AUX5) project.

**Institutional Review Board Statement:** Not applicable.

**Informed Consent Statement:** Not applicable.

**Data Availability Statement:** Not applicable.

**Conflicts of Interest:** The authors declare no conflict of interest.

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
