# Peer review of "The Bike-Sharing System in Slovakia and the Impact of COVID-19 on This Shared Mobility Service in a Selected City"

_sustainability, doi:10.3390/su13126544_

Round 1
Reviewer 1 Report
The authors studied effects of COVID-19 pandemic on bike-sharing system in Košice. Here are my questions/comments:
Introduction:
Introduction in written in a clear way, includes all necessary data, summarizes the literature and clearly states the authors aims. Nevertheless, in line 47, where disadvantages of bike sharing are listed, following disadvantages should be added:
- health issues (older people and people with bad physical fitness are discouraged to use bikes)
- Luggage (bike is not an adequate transportation for people carrying heavy or large luggage)
Section 2:
The aim of this section is not clear. The aim of the article is to describe the influence of COVID-19 on bike-sharing system. As the influence on COVID-19 in studied only in Košice, it is unclear, why the other bike sharing systems in Slovakia are mentioned. The section would have purpose if Košice would be compared to other bike-sharing system in more clear way considering the significance of such comparison to the contest of the article as a whole. Therefore I recommend authors to rewrite the section 2 in a way to omit all unnecessary data and to connect its content to content of sections that follow it.
Section 4:
In general, section includes a lot of data without a clear indication what authors wanted to present with them. On other hand, other data, that may be interesting or important, are missing. It gives the impression that authors simply presented data that they have and did not consider their purpose or put any effort in gathering additional data important for the analysis. Also, analysis and the discussion of data is poor and inadequate. Here are more detailed comments:
- Authors want to study influence of COVID-19 pandemic on bike sharing system. Data about bike sharing users are presented, but there are no data considering COVID-19. The authors should at least include data like time evolution of COVID-19 cases detected in a country as well as time-line of implementation of different government measures/restrictions.
- In line 303 authors state that no. of bike-sharing in 2020 in December are higher due to good weather conditions and revealing restrictions. This can not be seen from data as information of weather conditions and government COVID-19 restriction is missing.
- In many cases authors in a text state the max. and min. values of parameters in graph without the following discussion. As those data could be clearly seen from the graph, this is simply doubling of information. So it should either be omitted or the significance additional mentioning of min. and max. values should be explained.
- Please explain why are Figures 6., 7. 8. and 9. included. There is no indication that the habit of bike rentals have changed according to day of week or time of the day, so I do not have see any reason why a lot of attention is given to these data. I also see no reason why special attention is given to August as pattern does not seem to differ from other months. Either this should be explained or Fig. 8 and 9 should be omitted while Fig. 6. and 7. should be given less attention (just a comment that the day/hourly pattern did not change). It would be useful if author could
collect more data that can explain such a behavior considering the structure of users (people going to work, students going to university lectures, people going to cinema and night life….)
- Similar, I can not see any additional value in Fig. 10. There seems to be no change in pattern comparing time before and during COVID-19 pandemic (like that the change is evident more in some area of the city and less in others).
- I do not see any indication in the results that can lead to the conclusions given in statements in text from line 377 to the end of the section. Authors mention that the young residents and visitors are using a system, but this can not be proven from the data. Also it can not be seen from the data that most share points are located near cinemas, universities. Neither the data indicate how the number of bikes influences the usage of the system. This statements should be either omitted or supported with data.
Similar, all statement given in conclusion are not supported from the data given in article and have nothing to do with the COVID-19 situation (growing popularity, dense networks….)
In general, only two conclusions that can be drown from the data presented in the article are:
- The number of users of bike-sharing system has dropped significantly during COVID-19 crisis (as expected)
- The time rental is longer (but no reason for this was found)
In their analysis authors should also include at least data about COVID-19 crisis (government restrictions and number of positive cases evolution) as well as survey about structure of bike sharing system users (how many people use it for going to work, school, leisure…). Potentially other data can be included as well (e.g weather data, comparison with other transportation means). All conclusions have to be supported by the presented data.
Reviewer 2 Report
The paper provides a perspective on bicycle-sharing in Slovakia which is interesting, but perhaps not enough to justify publication. The full potential remains to be realized. The paper should be revised to better address the ‘so what?’ question. The paper would benefit from some statistical analysis of the data. The paper is comprehensible, but language editing is also needed.
References [1,2] on line 28 come after reference [3] on line 23.
Line 57 and line 65 both refer to ‘use of the bike-sharing system’.
The Introduction could be improved by stating the purpose of the study, highlighting the research question(s) and briefly explaining what follows.
In Table 3, please state what year the data is for and if the number of bicycle rentals is per year or other time period. How are the approximate values arrived at?
For the paragraph at line 196: There should be a better justification for selecting the most advantageous service. In Piestany, what is the estimated demand? At line 135 the number of inhabitants per 1 bicycle is used as ‘possible demand’, so that would assume that demand is being met. The population per bicycle is the lowest of all the cities. ‘All bicycles are often rented.’ in Table 5. What data is this statement based on? Please explain.
In Section 3 (line 202), Kosice is selected for analysis. An explanation for choosing this city should be provided.
Line 257 appears to be describing a literature review. The connection to processing the data should be explained (line 256). Which literature was relevant to the choice of data processing?
Lines 256-266 appear to be describing the methodology for the entire paper, not just the processing of the provided data as stated on line 256. On line 236 ‘methodology’ should read ‘methods’.
On line 288 ‘Part of our research…’ what was the other part? The purpose of the study is not clear. See comment above regarding the Introduction.
On line 292-3, it is stated that the decrease is caused by COVID-19. Justification should be provided. No historical data exists (before 2019). Could there have been other factors such as weather, economic decline, etc.? Similarly line 304 – cause and effect have not been rigorously established. It would also be useful to describe what COVID-19 restrictions were in place during the months in question.
Line 369-370: What justification is there for people’s fear causing the decrease?
Line 374: There is an increase, but is it statistically significant? The authors should include some statistical analyses e.g., two sample t-test, for this and other data presented (mean, standard deviation).
Line 377 – 395 is descriptive and general. The space should be used to discuss the implications of the results. For example, it is not clear why people would be afraid of using bike-sharing where they may meet other people outside (line 370) but this makes sense for public transport. This needs explanation. The interesting result (longer rental) is not connected to any reason. The discussion should discuss possible reasons for the interesting result.
Conclusion: The two purposes of the paper are finally clearly stated.
Describing the bike-sharing systems in cities in Slovakia is interesting, but the recommendations are not clearly connected with the data e.g., what analysis of the data indicates that a dense network should be implemented? Where have user expectations been investigated? Are there any overarching recommendations for the development of future bike-sharing systems based on the characteristics and features associated with the investigated bike-sharing systems supported by the data?
For the Covid-19 part, the data was compared for 2019 and 2020, but what conclusions can be drawn from the comparison e.g., people used bike-sharing less so what does this mean for the bike-sharing companies, public policy, public education, etc.?
Round 2
Reviewer 1 Report
Authors answered to all my question and included the changes that I proposed, so I recommend editors to accept the paper for publication.
Nevertheless, I advise authors to still add the following changes:
- It would be interesting to compare data in Fig. 7 and 6. From Fig. 7 one can get the data on reduction of using bike-sharing system in % which can be than compared directly with data on Fig. 6
- To rewrite the lines 873, 874. Two sentences start with "on the other hand", which sounds bad. Using some other word combination would sound better.
Reviewer 2 Report
The manuscript has been improved by the authors.